# Integrated analysis of long non-coding RNAs and mRNA profiles reveals potential sex-dependent biomarkers of bevacizumab/erlotinib response in advanced lung cancer

Chao Tu[1]ᴼ*, Yingqi Pi[2]ᴼ, Shan Xing[3], Yang Ling[4]*

1 Department of Internal Medicine, The Third Affiliated Hospital of Soochow University, Changzhou, Jiangsu, China, 2 Department of Clinical Laboratory, The Seventh Affiliated Hospital, Sun Yat-sen University, Shenzhen, China, 3 Department of Clinical Laboratory, State Key Laboratory of Oncology in South China, Collaborative Innovation Center for Cancer Medicine, Sun Yat-sen University Cancer Center, Guangzhou, China, 4 Department of Oncology, The Third Affiliated Hospital of Soochow University (Changzhou Tumor Hospital Affiliated to Soochow University), Changzhou, China

ᴼ These authors contributed equally to this work.
* tcmedical21@126.com (CT); 2559512439@qq.com (YL)

**Data Availability Statement:** All relevant data are within the manuscript and Supporting Information files.

## Abstract

### Background

While lung cancer patient outcomes are well-recognized to vary as a function of patient sex, there has been insufficient research regarding the relationship between patient sex and EGFR(Epidermal growth factor receptor) response efficacy. The present study therefore sought to identify novel sex-related biomarkers of bevacizumab/erlotinib (BE) responses in non-small cell lung cancer (NSCLC) patients.

### Methods

The exon array data in the Gene Expression Omnibus (GEO) dataset were analyzed in order to identify patterns of mRNA and lncRNA expression associated with BE resistance in NSCLC. These differentially expressed (DE) lncRNAs and mRNAs were identified via DE Analysis Filtering. These DE mRNAs were then assessed for their potential functional roles via pathway enrichment analyses, with overlapping functions possibly associated with the BE resistance. The mRNAs in these overlapping groups were then assessed for their correlations with patient survival, and lncRNA-mRNA co-expression networks were generated for each patient subset. A protein-protein interaction (PPI) network was also generated based upon these DE mRNAs.

### Results

In females we identified 172 DE lncRNAs and 1766 DE mRNAs associated with BE responses, while in males we identified 78 DE lncRNAs and 485 DE mRNAs associated with such responses. Based on the overlap between these two datasets, we identified a total of 37 GO functions and 18 pathways associated with BE responses. Co-expression

**Funding:** The author(s) received no specific funding for this work.

**Competing interests:** The authors have declared that no competing interests exist.

**Abbreviations:** BE, bevacizumab/erlotinib; DE, Differentially expressed; EGFR, epidermal growth factor receptor; NSCLC, non-small cell lung cancer; VEGF, vascular endothelial growth factor; lncRNAs, long noncoding RNAs; mRNA, Messenger RNA; GEO, Gene Expression Omnibus; DE lncRNAs, differentially expressed lncRNAs; GO, Gene Ontology; KEGG, Kyoto Encyclopedia of Genes and Genomes; DAVID, Database of Annotation Visualization and Integrated Discovery; PPI network, protein protein interaction network; LUSC, lung squamous cell carcinoma; LUAC, lung adenocarcinoma; TCGA, The Cancer Genome Atlas; PI3Ks, Phosphoinositide 3-kinases; CDH(5), CDH5 Recombinant Cadherin 5; ERG, Endoplasmic reticulum Golgi.

and PPI networks suggested that the key lncRNAs and mRNAs associated with these BE response mechanisms weredifferent in the male and female patients.

## Conclusions

This work is the first to conduct a global profiling of the relationship between lncRNA and mRNA expression patterns, patient sex, and BE responses in individuals suffering from NSCLC. Together these results suggest that the integrative lncRNA-mRNA expression analyses may offer invaluable new therapeutic insights that can guide the tailored treatment of lung cancer in order to ensure optimal BE responses.

## Introduction

Lung cancer is a leading driver of cancer-associated death globally [1], with most patients only being diagnosed when the disease is in its advanced stages and is no longer eligible to undergo curative surgical resection [2]. In non-small cell lung cancer (NSCLC), epidermal growth factor receptor (EGFR) mutations are frequently detected [3]. Erlotinib was developed as an inhibitor of EGFR to treat patients with NSCLC that have activating mutations in EGFR [4–6], but how effective this compound is in patients with wild type EGFR remains less clear. Some initial studies have suggested that firstline treatment with erlotinib or a similar kinase inhibitor in such patients may even be more harmful that conventional chemotherapy [7], although more recent biomarker analysis of large-scale trial found that a subset of EGFR wild-type patients do benefit from such treatment [8].

A leading therapeutic strategy for treating cancer revolves around the simultaneous targeting of multiple targets via a combination treatment approach in an effort to simultaneously suppress multiple signaling pathways within tumor cells [9, 10]. The combination of erlotinib with the bevacizumab, which is an antibody specific for vascular endothelial growth factor (VEGF) has been found to be more effective than treatment with erlotinib alone [11]. While EGFR activating mutations are the biomarkers most closely associated with patient responses to this combined bevacizumab + erlotinib (BE) treatment, certain patients without such mutations have also been found to benefit from combination therapy. Previous studies have investigated the efficacy and safety of erlotinib plus bevacizumab treatment in patients with non-small-cell lung cancer [12, 13]. However, the exact molecular mechanisms underlying BE responsiveness in these patients remain uncertain. It is therefore vital that additional biomarkers of BE responses be identified in order to more effectively identify those EGFR wild-type patients who are most likely to benefit from BE treatment in order to guide appropriate patient treatment strategies.

Solid tumor biomarkers and therapeutic targets have increasingly been identified using omics technologies [14], with whole-genome exon arrays being ideal for discovering such biomarkers with exon-level resolution [15]. These arrays permit for gene- and exon-level analysis of the expression of both mRNAs and long non-coding RNAs (lncRNAs). These lncRNAs are known to be capable of regulating a variety of activities including RNA splicing, stability, and shuttling between the nucleus and the cytoplasm [16]. The integrated analysis of both mRNA and lncRNA expression patterns in tumor cells has the potential to offer invaluable insight into the molecular basis for tumor pathogenesis, thereby potentially identifying prognostic or therapeutic targets to guide patient treatment efforts. To date, however, no studies have

constructed a lncRNA-mRNA interaction network specifically associated with BE therapeutic responses in lung cancer.

There is clear evidence that men typically have poorer lung cancer outcomes than do women [17]. There is additionally evidence suggesting that there are sex-specific differences in immunological functionality and in patient responses to targeted molecular therapies. We therefore hypothesized that such sexual dimorphism may have an impact on the relative efficacy of BE responses in male and female lung cancer patients.

In the present study, we analyzed patient subgroups from among individuals participating in the EGFR wild-type group of the SAKK 19/05 trial in an effort to better understand what factors influence patient responses to BE therapy. In particular, we focused on the relationship between patient sex and genetic factors as they related to BE responses in this patient cohort. For this analysis, we analyzed published whole-exome array data available in the Gene Expression Omnibus (GEO) in order to identify those mRNAs and lncRNAs that were differentially expressed (DE) between patients that did and did not respond to BE therapy in male and female patient subgroups. When then constructed separate lncRNA-mRNA coexpression networks pertaining to BE responses in males and females separately, and analyzed the similarities and differences between the networks in these two groups. Together, the results of our analysis highlight that there are sex-dependent differences in patient responses to targeted molecular therapies [18–20], suggesting that such integrative lncRNA/mRNA-based analyses may help to better predict patient therapeutic responses, thereby allowing clinicians to select appropriate treatments for lung cancer patients in an individually targeted manner.

## Materials and methods

### Data sets

We obtained publicly available gene expression profile data (GSE37138) from GEO (http://www.ncbi.nlm.nih.gov/geo) for the present analysis. This dataset included baseline samples obtained from the whole bronchoscopic biopsy of 42 suffering from advanced non-squamous NSCLC. These patients had then undergone treatment with BE either until disease progression was observed or until toxicity necessitated the termination of this treatment. Based on whether or not they exhibited stable disease after 12 weeks of treatment, patients were categorized as either responders (n = 22; 12 male, 10 female) or non-responders (n = 20; 12 male, 8 female). Additional detail regarding the SAKK 19/05 trial is discussed in published articles [21]. Data and probe sequences were analyzed in accordance with the GPL5188 [HuEx-1_0-st] Affymetrix Human Exon 1.0 ST Array, with background signals being adjusted according to the Robust Multichip Average.

### mRNA and lncRNA profiling

Using the provided exon array data, we sought to assess the mRNA and lncRNA expression profiles in these patient samples. To that end, we began by constructing an mRNA and lncRNA alignment database. The exon array enrichment uniquely mapped to genes and the lncRNAs were indeed inferred from genes. In order to identify mRNAs, those transcripts that were preceded by either NM or XM in the NCBI refseq database and those transcripts annotated to code for protein in the Ensembl database (http://www.genome.ucsc.edu/) were pooled together, with any repeats being removed. In order to identify lncRNAs, those RNAs preceded by either "NR_" or "XR_" in the Refseq database, those that were annotated with "lincRNA", "processed transcripts", "non-coding", or "misc_RNA" in the Ensembl database, and those that were found within the NONCODEv5 database were pooled together. After removal of repeats, those non-coding RNA sequences that were > 199 bp long were then selected to yield

a lncRNA alignment database. Next, BLAST (https://blast.ncbi.nlm.nih.gov/Blast.cgi) was carried out with the bidirectional best hit (BBH) criteria to map the 141,1080 probe set IDs for this dataset in order to yield an mRNA alignment database with an e-value threshold of 10e-5. Those probe sets aligned with these mRNAs were annotated as such, with the remaining probes being mapped to the lncRNA alignment database. Only those matched lncRNAs with an e-value threshold of 10e-5 were included in this analysis.

### Identification of mRNAs and lncRNAs associated with BE responses

Those mRNAs and lncRNAs that were differentially expressed between responders and non-responders in both the male and female patient subgroups were next identified. These DE lncRNAs and DE mRNAs were identified via DE Analysis filtering, using filtering criteria of fold change >2 and $p < 0.05$ and fold change >1.2 and $p < 0.05$, respectively.

### Functional enrichment analysis

DE mRNAs were separated into four groups according to the sex and direction in which they were differentially regulated (male-upregulated; male-downregulated; female-upregulated; female-downregulated). The DE mRNAs in these groups then underwent both Gene Ontology (GO: http://www.geneontology.org) and KEGG (http://www.genome.ad.jp/kegg/) pathway analyses using DAVID (http://david.abcc.ncifcrf.gov/). For the GO analyses, the biological processes associated with the input mRNAs in an annotated database were used to identify regulatory gene networks based upon the hierarchical relationships between these genes, while KEGG analyses offered insight into the pathways in which these DE mRNAs participate. Only those GO and KEGG terms meeting the following criteria were included in these analyses: $p<0.001$, FDR$<0.05$ [22].

We reasoned that those functional pathways that were selectively enriched in both male and female NSCLC patients may represent the pathways most closely associated with BE responses. We therefore compared such shared pathways between these two sex groups in order to assess whether the same genetic profiles were responsible or these BE response-associated phenotypes in both males and females. The same analysis was used to compare both up- and down-regulated KEGG and GO terms between male and female patients.

### lncRNA-mRNA correlation network

Cytoscape (http://www.cytoscape.org) was used to generate two lncRNA-mRNA correlation networks (one each for males and females) based upon the predicted targets of DE lncRNAs and the available DE mRNA data. Normalized signal intensity values for DE lncRNAs and DE mRNAs, as generated above, was used for network construction. In addition, as most lncRNAs included in this analysis have not been functionally annotated, we conducted a co-expression analysis in order to identify those mRNAs that vary their expression in a pattern closely related to the expression pattern for a given lncRNA. Such analyses offer potentially insight into the poorly understood roles for these lncRNAs as regulators of protein-coding gene expression. The resultant co-expression network figure makes note of both the directionality of lncRNA/mRNA differential expression, as well as associated GO and KEGG terms and the correlative relationships between different transcripts. In addition, degree centrality was determined based upon the number of links between nodes, serving as an indicator of the relative biological importance of a given transcript.

## PPI network analysis

The STRING database was used to screen for known or predicted interactions between proteins of interest in the present study, with a specific focus on interacting protein pairs associated with DE mRNAs in the lncRNA-mRNA correlation network. Only those interactions with a confidence cutoff > 0.4 were included in the resultant network, which was visualized using Cytoscape.

## Survival analysis

We additionally examined the relationship between specific DE mRNAs identified in the present study and patient survival outcomes. To that end, the Cancer Genome Atlas (TCGA) was used to identify samples from LUSC (lung squamous cell carcinoma) and LUAC (lung adenocarcinoma) patients for which clinical follow-up data was available. Patients in these cohorts were then divided into two groups based on whether they expressed high or low levels of a given gene of interest, and the overall survival (OS) between these two groups was then compared via the Kaplan-Meier method, with log-rank tests used to gauge significance.

## Statistical analyses

The Bioconductor limma package (v3.26.1) and R studio (v3.2.2) were used to identify DE lncRNAs and DE mRNAs between BE-responsive and -unresponsive patient samples in the present study. FDR values were used to control for multiple testing. $P<0.05$ was the significance threshold.

# Results

## Patient characteristics and clinical outcomes

We began by accessing the GSE37138 dataset containing exon array data from 42 NSCLC patients that had undergone targeted BE treatment. These patients were from 35–78 years old (median: 61), with all being of good performance status (WHO 0–1). Of these patients, 24 were male (57.1%) and 18 were female (42.8%), with almost all patients (92.9%) having stage IV disease. EGFR and KRAS mutations were detected in 4 (9.5%) and 6 (14.3%) of patients, respectively.

## Identification of differentially expressed lncRNAs and mRNAs

We next compared the expression of lncRNAs and mRNAs in both male and female patients in this dataset in order to identify those transcripts that were differentially expressed as a function of patient BE response. In females, we identified 172 DE lncRNAs (113 up- and 59 down-regulated) and 1766 DE mRNAs (866 up- and 900 down-regulated), while in males we identified 78 DE lncRNAs (44 up- and 34 down-regulated) and 485 DE mRNAs (179 up- and 306 down-regulated). These data were then organized into both volcano plots (Fig 1) and were visualized following hierarchical clustering into a heatmap (Fig 2).

## Functional enrichment analyses of DE mRNAs

We next conducted GO- and KEGG-based functional enrichment analysis of the DE mRNAs identified above in order to understand the relationship between these genes and BE therapeutic responses. These DE mRNAs were analyzed separately for both males and females.

In the female dataset up-regulated DE mRNAs were enriched for 747 GO terms, including signal transduction (Fig 3A), while the most enriched term corresponding to down-regulated

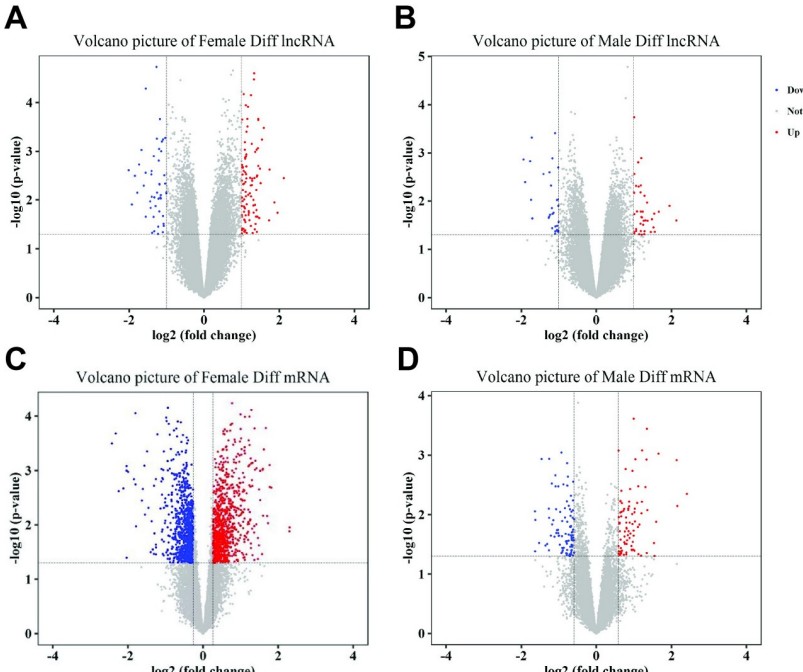

**Fig 1. Differential expression of mRNAs and lncRNAs in NSCLC patients as a function of BE responses.** (A) Differences in lncRNA expression in female BE-responsive patients; (B) Differences in lncRNA expression in male BE-responsive patients; (C) Differences in mRNA expression in female BE-responsive patients; (D) Differences in mRNA expression in male BE-responsive patients. The color scale corresponds to relative transcript expression, with green and red corresponding to low and high expression, respectively. BE: bevacizumab/erlotinib.

mRNAs was 'apoptotic process' (Fig 3B). In males, up-regulated mRNAs were most closely associated with nucleosome assembly, positive regulation of transcription from RNA, transcription from RNA polymerase II promoter, chromatin organization, and positive regulation of transcription, and DNA-templates (Fig 3C), while the most enriched term corresponding to down-regulated mRNAs was neutrophil degranulation (Fig 3D).

A pathway enrichment analysis revealed that 150 annotated pathways were associated with upregulated DE mRNAs in the female dataset, with the 'Ras signaling pathway' being the most enriched (Fig 4A). In addition, 102 annotated pathways were associated with downregulated DE mRNAs in the female dataset, with 'metabolic pathways' being the most enriched (Fig 4B). In contrast, upregulated DE mRNAs from the male dataset were enriched in 16 pathways, of which 'Systemic lupus erythematosus' was the most strongly enriched (Fig 4C), while downregulated DE mRNAs from the male dataset were enriched in 67 pathways, of which 'metabolic pathways' were also the most enriched (Fig 4D).

In order to identify those enriched GO terms and pathways most closely associated with BE responses, we identified overlapping terms between the male and female datasets, revealing 11 and 26 shared GO functions that were up- and down-regulated, respectively, and 4 and 14 pathways that were up- and down-regulated, respectively in both males and females. These overlapping pathways, including 'apoptotic process', 'metabolic pathways', 'PI3K-Akt signaling', and 'pathways in cancer' are thus likely associated with BE responsiveness (S1 and S2 Tables, and Fig 5). Among them, the positive/negative regulation of transcription from RNA polymerase II promoterGO functions and metabolic pathways enriched in both up-regulated and down-regulated groups. Despite these similarities with respect to GO and KEGG terms between male and female datasets, however, the genes which gave rise to these overlapping

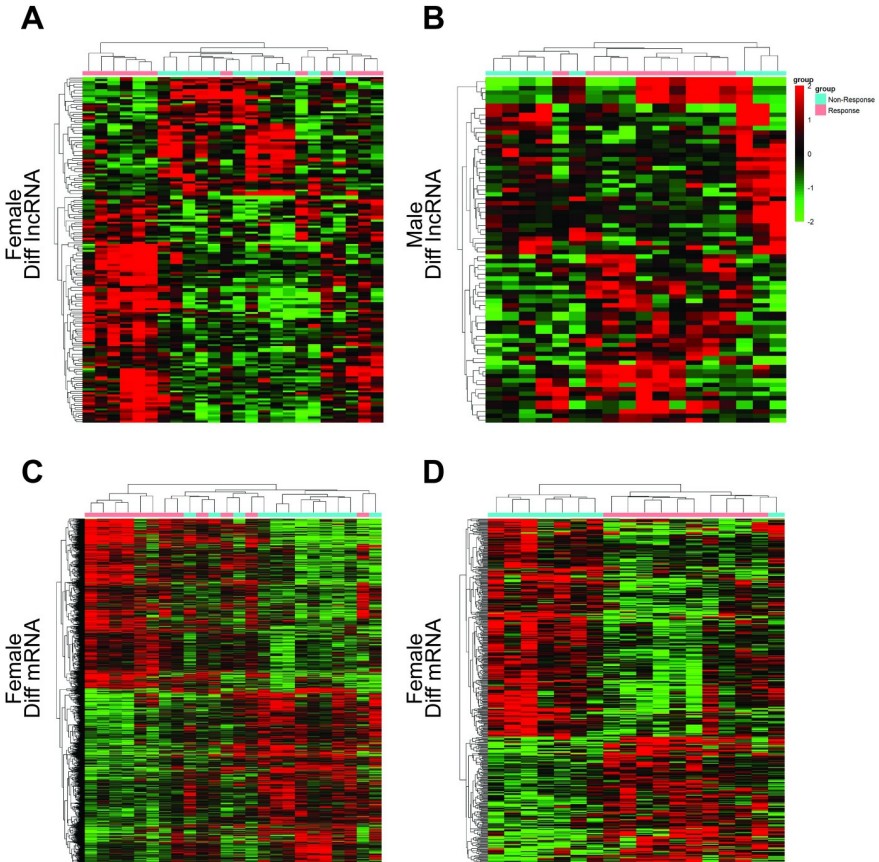

**Fig 2. Hierarchical clustering analysis of differentially expressed lncRNAs and mRNAs.** (A) Differences in lncRNA expression in female BE-responsive patients; (B) Differences in lncRNA expression in male BE-responsive patients; (C) Differences in mRNA expression in female BE-responsive patients; (D) Differences in mRNA expression in male BE-responsive patients. The color scale corresponds to relative transcript expression, with green and red corresponding to low and high expression, respectively. BE: bevacizumab/erlotinib.

similarities differed in males and females. In the female group, 433 DE mRNAs corresponded to the up-regulated GO function type, 542 DE mRNAs were involved in the down-regulated GO terms and 191 DE mRNAs in both up- and down-regulated GO functions, whereas in males the numbers were 124, 262 and 69, respectively. Similarly, in females 83 and 227 DE mRNAs were associated with the up- and down-regulated KEGG pathways, respectively, whereas in males these numbers were 47 and 109, respectively. In addition, 132 mRNAs in females and 53 in males were included in both up- and down-regulated pathways. These mRNAs were considered to be likely to play roles in regulating patient responses to BE therapy, and so were pooled together for subsequent analyses, yielding 256 DE mRNAs in males and 690 in females (929 total).

## Co-expression of lncRNAs-mRNAs in BE response lung cancer patients

We next constructed a lncRNA-mRNA correlation network in order to better understand how lncRNAs influence BE responses in NSCLC patients, incorporating both the DE lncRNAs and BE response-associated mRNAS identified above into the resultant network (Fig 6A and 6B). In total, the female network was made up of 282 nodes (232 mRNAs and 50 lncRNAs) and 458 connections, while the male network was made up of 202 nodes (177 mRNAs and 25

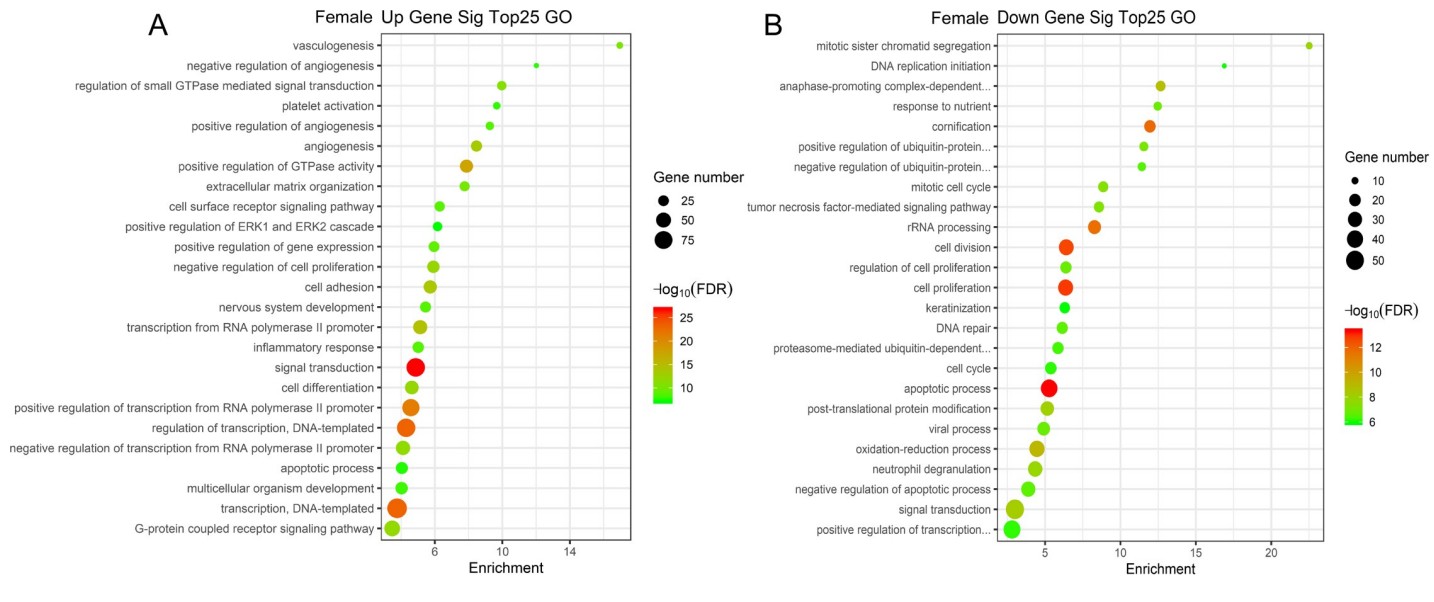

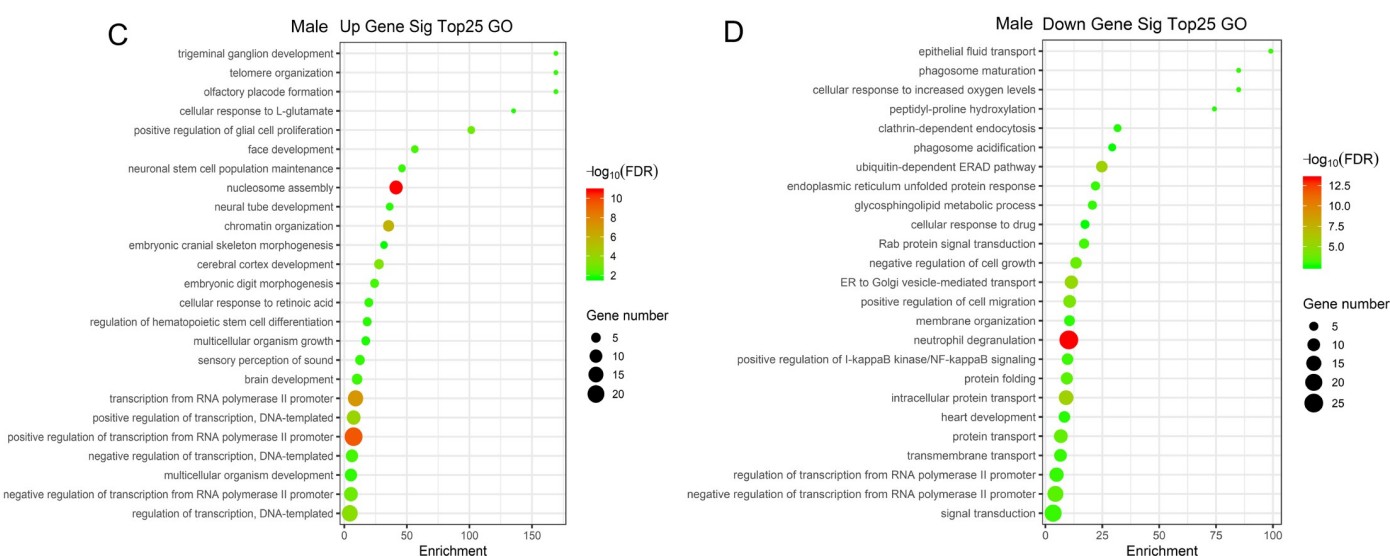

**Fig 3. The 25 most-enriched GO terms associated with BE response-related differentially expressed mRNAs in lung cancer patients.** (A) Enriched GO terms associated with upregulated mRNAs in females; (B) Enriched GO terms associated with downregulated mRNAs in females; (C) Enriched GO terms associated with upregulated mRNAs in males; (D) Enriched GO terms associated with downregulated mRNAs in males; BE: bevacizumab/erlotinib.

lncRNAs) and 375 connections. The majority of the co-expressed lncRNAs and mRNAs were positively correlated with one another. In many cases, individual lncRNAs were co-expressed with multiple mRNAs, suggesting the potential for a complex regulatory relationship between these molecules. In order to gauge the relative importance of the lncRNAs and mRNAs in these networks, the degree of centrality was calculated (S3 and S4 Tables). The nodes that possess the highest degree of centrality in females were: TEK, CDH5, ERG, ASPA, LDB2, TIE1, ACVRL1 and EDNRB, while in males the nodes with the highest degree of centrality were: SNX27, BLZF1, ATP6V1A, CHML, STAM, ACOX1, UBQLN1 and CRK. These results thus

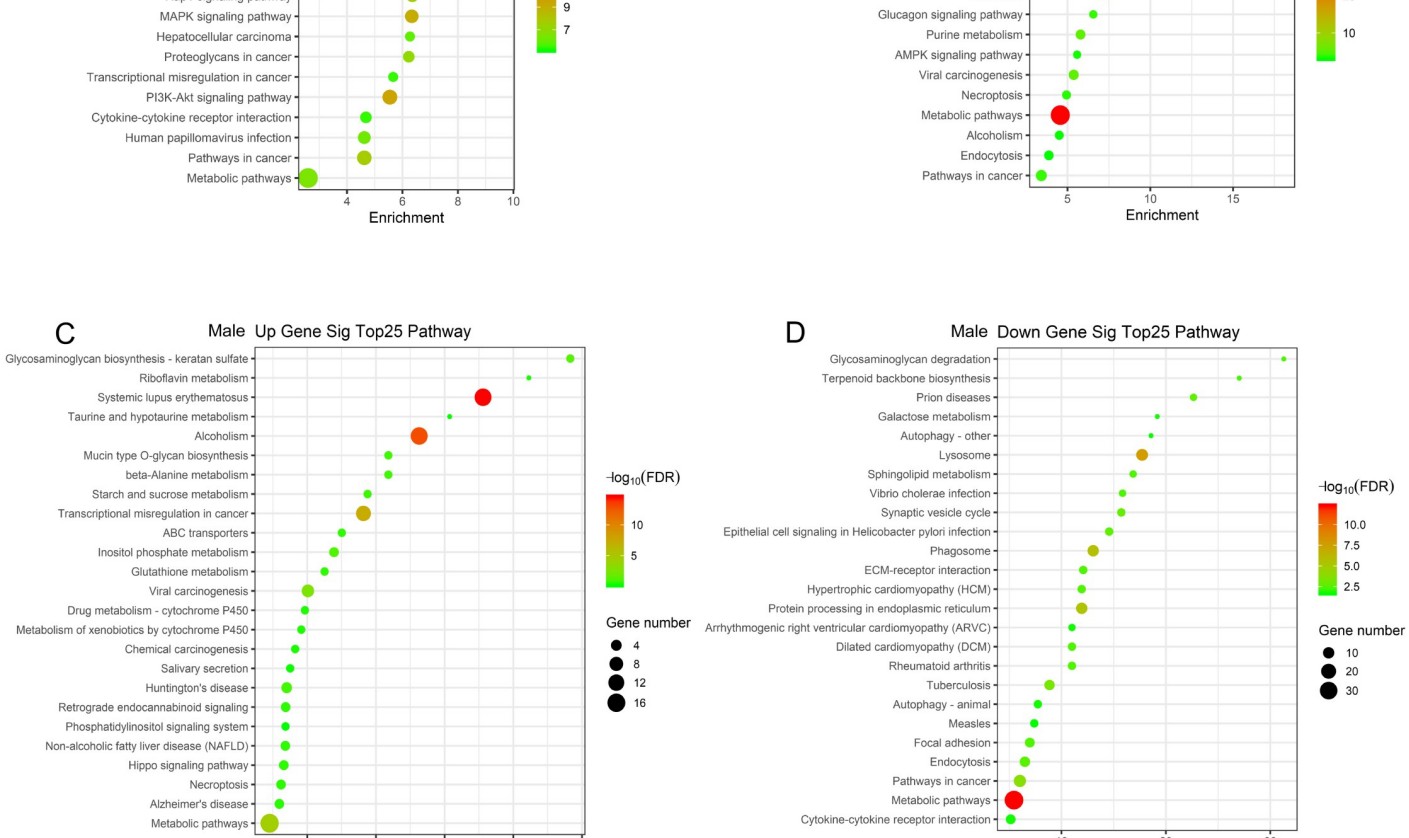

**Fig 4. The 25 most-enriched pathways associated with BE response-related differentially expressed mRNAs in lung cancer patients.** (A) Enriched pathways associated with upregulatedmRNAs in females; (B) Enriched pathways associated with downregulated mRNAs in females; (C) Enriched pathways associated with upregulated mRNAs in males; (D) Enriched pathways associated with downregulated mRNAs in males; BE: bevacizumab/erlotinib.

confirmed that there were sex-dependent differences in lncRNA-mRNA regulatory relation-ships in NSCLC patient tumor samples.

We further conducted the functional annotation of these co-expressed mRNAs in an effort to better understand lncRNA functionality (Fig 6C and 6D). The majority of lncRNA-associ-ated mRNAs in this network were associated with GO functions relating to metabolic path-ways and the positive regulation of transcription from RNA polymerase II. The key hub genes identified through these analyses differed between groups, with female hub genes including TEK, ERG, NONHSAT095695.2, LDB2, ENST00000552253.1, ENST00000588495.5, CDH5, and MEF2C, and male hub genes including ENST00000460756.5, G6PC3, ATF6, BMP2, SIX1,

A — Female&Male Shared Sig GO

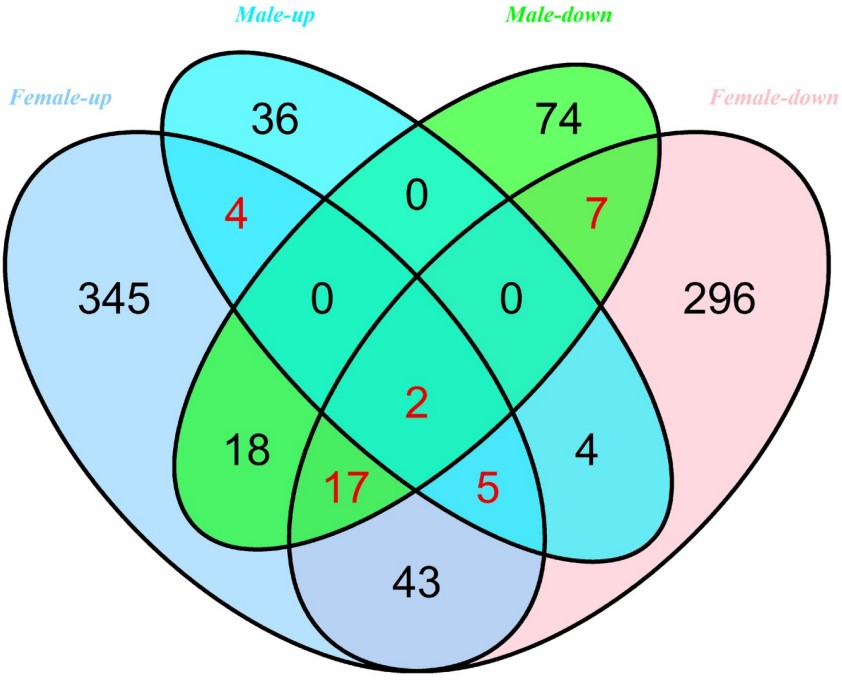

B — Female&Male Shared Sig Pathway

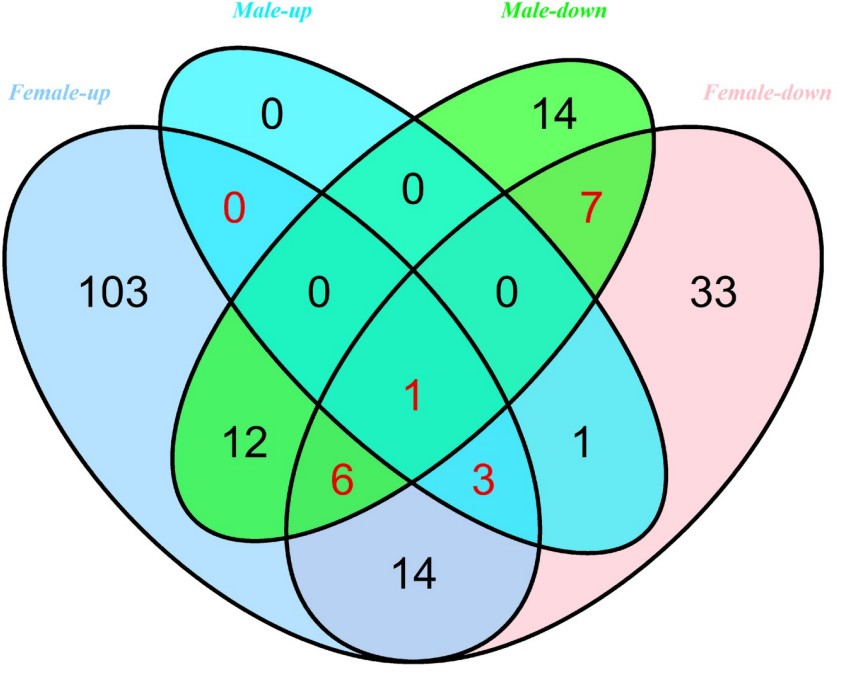

**Fig 5. Venn diagram analysis comparison of functional enrichment analysis results.** Significantly enriched (A) GO terms and (B) pathways overlapping between the female-upregulated, male-upregulated, female-downregulated, and male-downregulated DE mRNA datasets.

RAB7A, ENST00000575695.5, and SOX2 (S5 and S6 Tables). NONHSAT095695.2, ENST00000552253.1, and ENST00000588495.5 primarily regulated LDB2, ENST00000460756.5 primarily regulated ATF6, and ENST00000575695.5 primarily regulated UBQLN1. Together this network analysis thus highlights the potential for sex-specific differences in the drug response mechanisms in NSCLC patients.

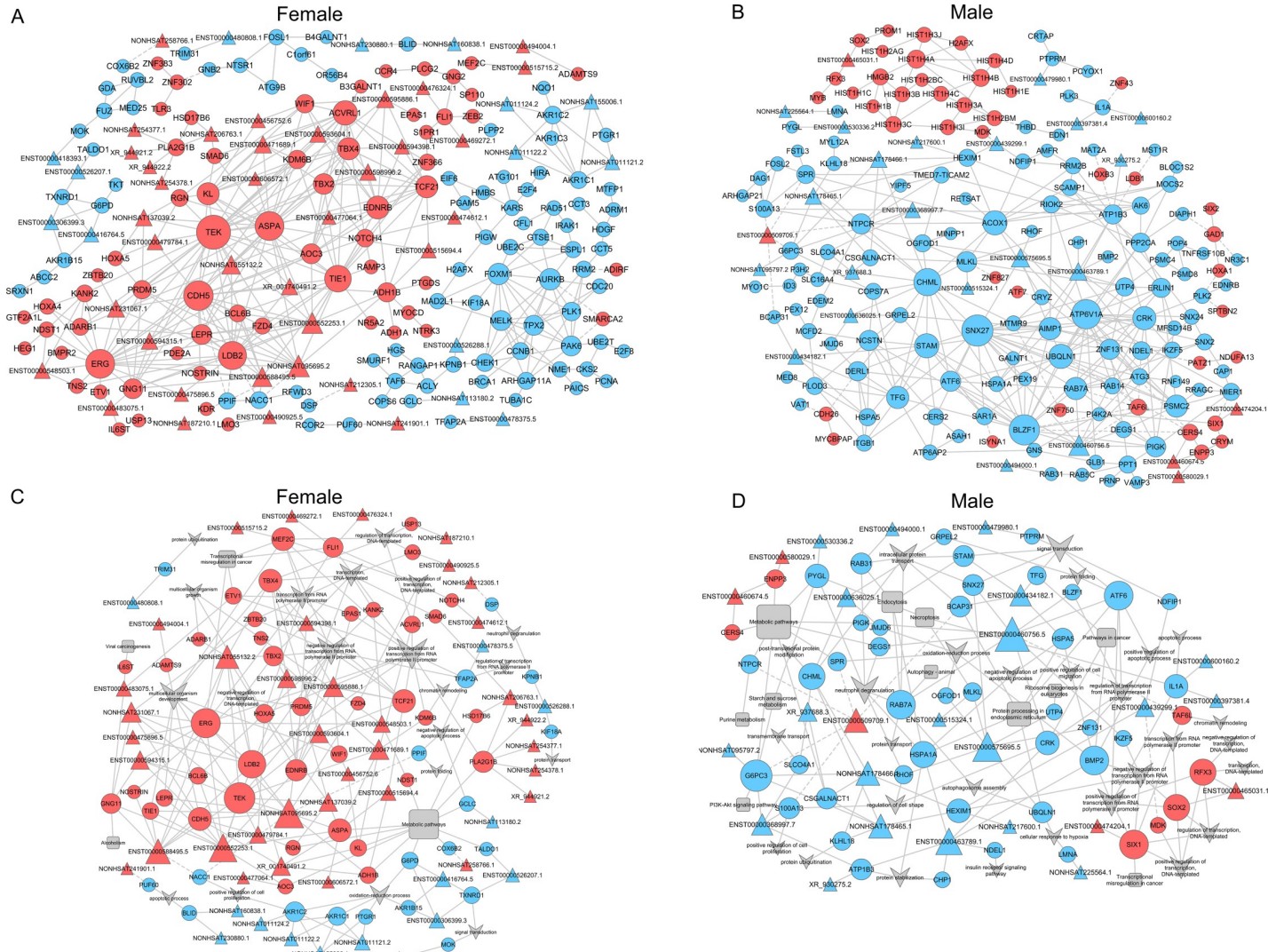

**Fig 6. Co-expression analysis of core mRNAs and correlated lncRNAs.** (A) A lncRNA-mRNA expression correlation network in females. (B) A lncRNA-mRNA expression correlation network in males. (C) A lncRNA-mRNA-GO-pathway correlation network in females. (D) A lncRNA-mRNA-GO-pathway correlation network in males. lncRNAs are represented by triangular nodes, while mRNAs are represented by circular nodes. GO terms are indicated by V-shaped arrow, while pathways are represented with rectangles. Red, blue, and gray nodes indicate upregulation, downregulation, or no significant regulation, respectively. Solid lines indicate positive direct correlations, while dotted lines correspond to negative correlations.

## PPI network construction

In an additional effort to understand these sex-dependent differences in BE responses, we generated a PPI network for the male and female patient cohorts, incorporating DE mRNAs that had been incorporated into the above lncRNA-mRNA co-expression network. As shown in Fig 7, TEK, LDB2, UBQLN1, and ATF6 all appeared to play key hub roles in their corresponding sex groups.

## Survival analysis

Lastly, we assessed whether the 929 DE mRNAs identified in our functional enrichment analyses were associated with different survival outcomes in LUAD and LUSC patients. To that end, a Kaplan-Meier-based approach was used to assess the prognostic value of these mRNAs in TCGA datasets. In total, we found that 266 of these mRNAs were significantly associated with LUAD patient survival, while 73 were associated with LUSC patient survival. In the LUSC patient cohort, 22 and 91 mRNAs were significantly associated with survival outcomes in females and males, respectively, while in LUAD 129 and 158 mRNAs were significantly associated with OS in females and males, respectively. Of the hub genes identified in females, we found that elevated expression of TEK, which was expressed at higher levels in biopsies from female BE-responsive patients, was correlated with better LUAD patient survival outcomes in females but not in males (Fig 8A and 8B). In contrast, decreased expression of UBQLN1, which was downregulated in biopsies from male BE-responsive patients, was associated with increased survival in male or total LUAD patients, but was unrelated to survival in female LUAD and LUSC patients (Fig 8C and 8D). Together these results highlight the potential relevance of sex-dependent mRNA expression profiles and corresponding differences in drug responses.

## Discussion

In the present study, we established profiles of differential mRNA and lncRNA expression that were associated with EGFR wild-type NSCLC patient responses to BE combination therapy. We found that profiles of DE mRNAs and lncRNA differed between male and female patients, although there was clear overlap with respect to the molecular pathways associated with BE responses in both patient groups. We further conducted lncRNA-mRNA co-expression analyses that highlighted sex-dependent heterogeneity with respect to the differential patterns of gene expression underlying patient responses. Survival analyses further confirmed that many of the hub genes identified in these analyses were associated with sex-dependent differences in lung cancer patient survival outcomes.

About 10% of Caucasian NSCLC patients and 40–50% of East Asian NSCLC patients exhibit EGFR mutations [23]. EGFR tyrosine kinase inhibitors (TKIs) have been shown to offer therapeutic benefit even in patients that do no exhibit such activating EGFR mutations [24–26], with a subset of EGFR wild-type patients exhibiting clinical responses to these treatments. The relevance of many biomarkers has been investigated in these EGFR wild-type patients, such as the expression of EGFR, EGFR ligand, an amplification of EGFR, all of which are associated with TKI responses [27]. However, there are currently no clear biomarkers that are suitable for the identification of NSCLC patients that are likely to respond well to EGFR TKI treatment, particularly in EGFR wild-type patients.

In this study assess the relationships between lncRNA and mRNA expression in exon arrays from NSCLC patients in order to explore genetic signatures associated with responses to a combination of an EGFR-TKI and an anti-angiogenic agent. However, more potential differentially expressed long non-coding RNAs and coding genes can be discovered if RNA-seq data

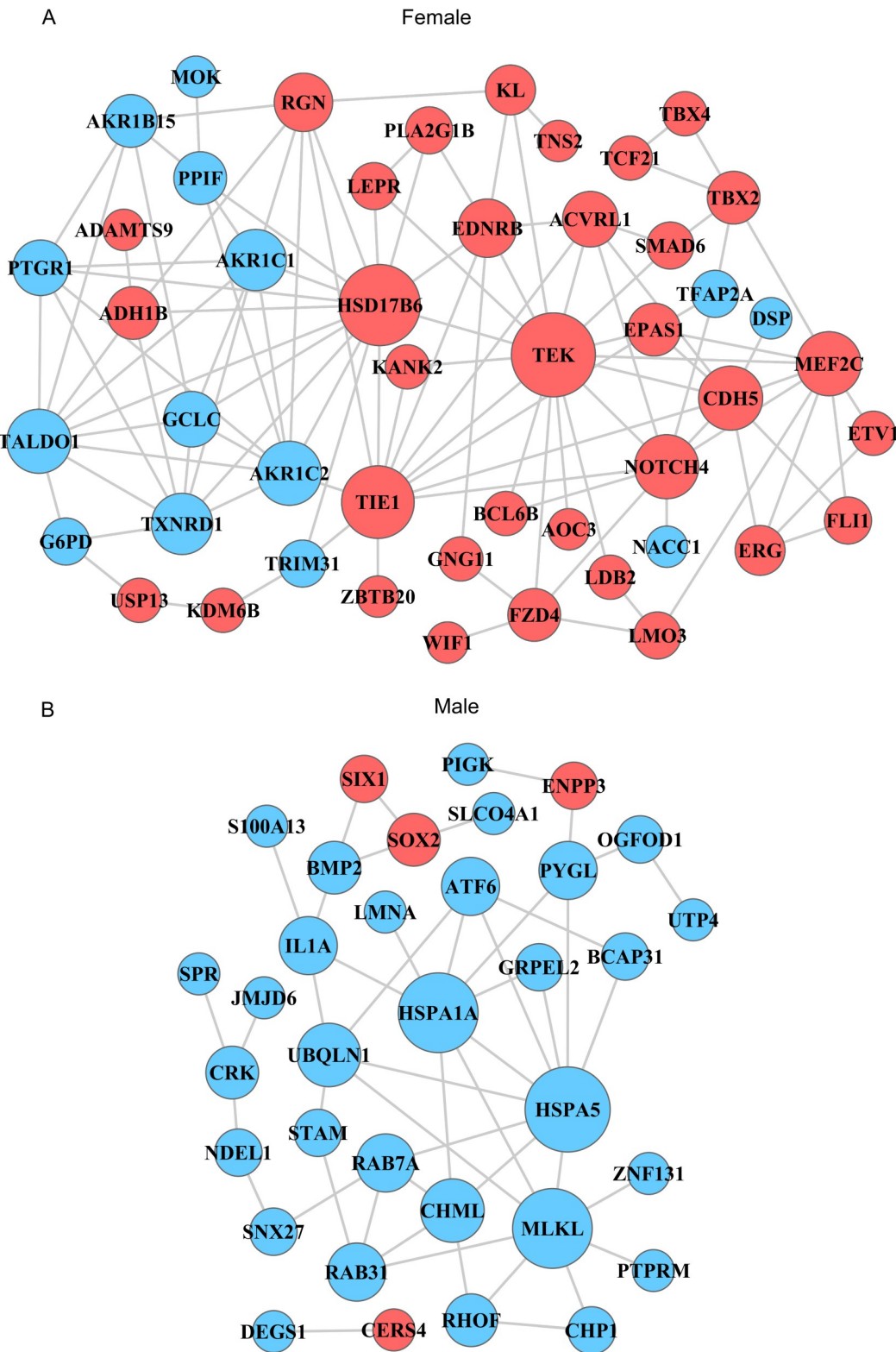

**Fig 7. Protein-protein interaction network containing for differentially expressed mRNAs in the lncRNA-mRNA correlation network.** Proteins are represented by nodes, with edges corresponding to protein-protein interactions. Upregulated and downregulated DE mRNAs are represented in red and blue, respectively. DE mRNAs: differentially expressed mRNAs.

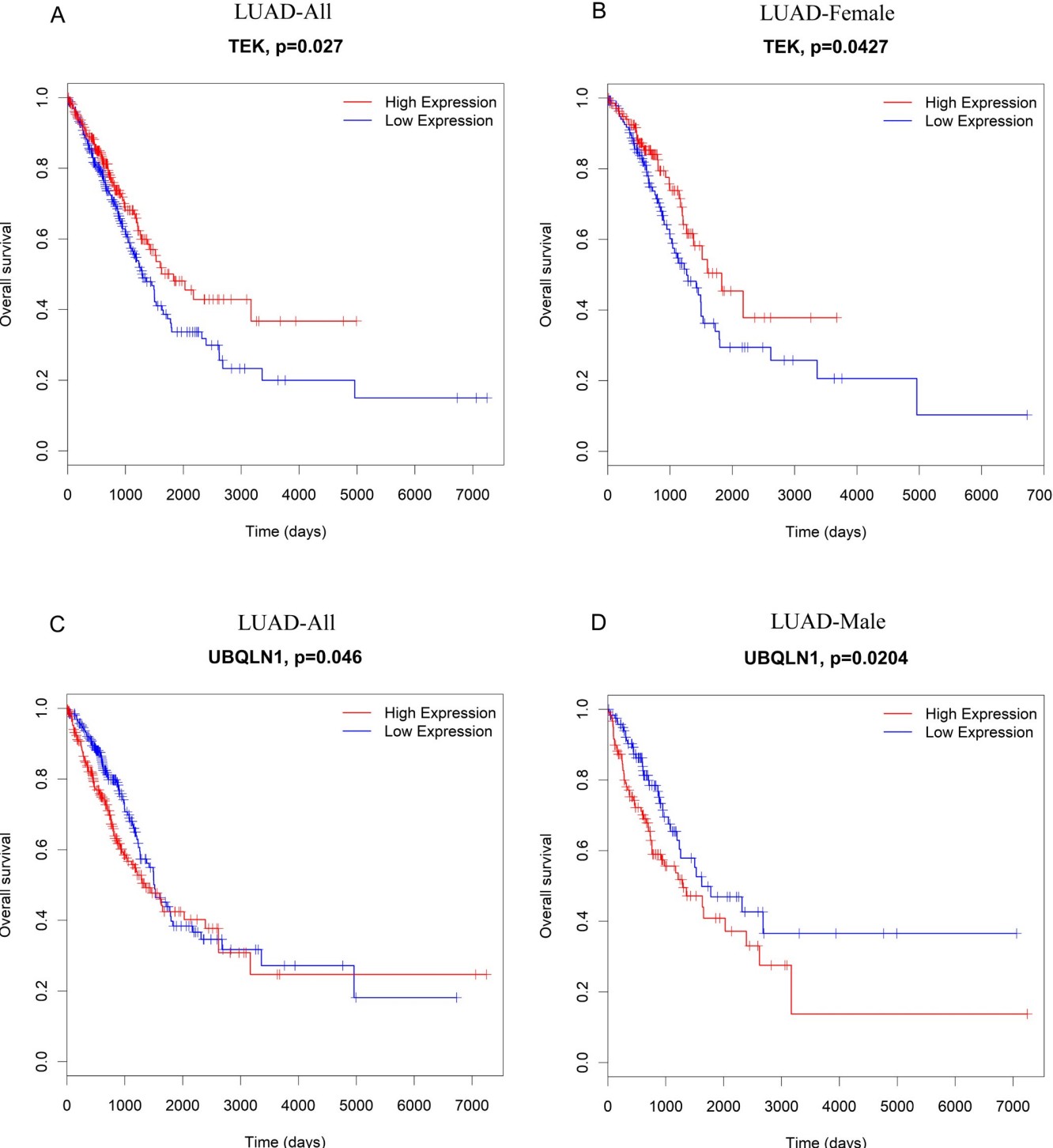

**Fig 8. The relationship between TEK and UBQLN1 expression and LUAD patient overall survival.** Kaplan-Meier survival curves were used to assess (A) survival as a function of TEK expression in all LUAD patients; (B) survival as a function of TEK expression in female LUAD patients; (C) survival as a function of UBQLN1 expression in all LUAD patients; (D) survival as a function of UBQLN1 expression in male LUAD patients. LUAD: Lung Adenocarcinoma.

is used instead of exon array, we only used exon array data in this work due to the data availability. In total we identified 37 GO functions and 18 pathways that were putatively linked to

BE responses, with prominent enriched functions including apoptotic processes, metabolic pathways, cancer-related pathways, and transcription-associated processes. These results suggested that these pathways are closely associated with lung cancer patient BE responses. Previous work has clearly demonstrated roles for these pathways in modulating cancer cell signal transduction, thereby altering the proliferation, growth, survival, and metabolism of tumor cells [28–30]. In a lncRNA-mRNA-GO-Pathway-network analysis, we further demonstrated the interrelatedness of these signaling pathways, highlighting a molecular network governing BE responses that were consistent with past experimental research [31].

Interestingly, we found that distinct lncRNAs and mRNAs were associated with BE responses in our male and female patient cohorts, with TEK, ERG, NONHSAT095695.2, LDB2, ENST00000552253.1, ENST00000588495.5, CDH5 and MEF2C being tightly linked to BE responses in females, and ENST00000460756.5, G6PC3, ATF6, BMP2, SIX1, RAB7A, ENST00000575695.5, and SOX2 being associated with these responses in males. In females, these core genes were upregulated in BE-responsive patients, whereas in males all of these core genes other than SIX1 and SOX2 were downregulated in BE-responsive patients. These results are of interest as they highlight sex-dependent differences in patient tumor responses, consistent with previous work demonstrating sexual dimorphism with respect to cancer patient physiological, pathophysiological, and pharmacological outcomes [32]. Work from the Moffitt Cancer Center indicated that men were associated with worse lung cancer outcomes than women (HR, 1.24 [95% CI, 1.09–1.41]) [33], and sex-related differences in patient immunotherapy have also been observed [34]. Indeed, immune checkpoint inhibitors have been shown to bolster the OS of advanced NSCLC patients in a sex-dependent manner [35]. Women have also been found to exhibit higher rates of EGFR mutations, thus making them more likely to respond to TKI treatment [7]. Work by Seo Yun Kim study 66 LUAC patients suggests that female EGFR wild-type patients are more likely to respond to EGFR TKI (odds ratio [OR], 3.10; 95% CI, 1.05–9.19; p = 0.041) than are males [36]. The specific mechanisms underlying these sex-dependent differences in pharmacological and physiological outcomes in men and women are, however, not well understood.

In this study we generated a lncRNA-mRNA co-expression network which allowed us to highlight male- and female-specific transcriptional signatures associated with BE responses. All of these hub genes have been linked with tumor growth and development, although they have not all been associated with drug responses. The TEK hub gene identified in females has been linked with pathologic complete responses to Bevacizumab in individuals with breast cancer [37]. Work by Song et al. further suggests that ERG is a biomarker predictive of docetaxel responsiveness in those with metastatic castration-resistant prostate cancer [38]. NONHSAT095695.2, ENST00000552253.1 and ENST00000588495.5 regulated LDB2, which inhibits liver cancer cell proliferation and migration via suppressing the expression of HEY1 [39].

Work by Xiaoman indicates that CDH5 can suppress the metastasis of breast cancer owing to the ability of its 3'-UTR serve as a ceRNA for STARD13 [40]. In addition, MEF2C has previously been identified as a possible leukemia-related oncogene [41].

In males, we found that ENST00000460756.5 primarily regulated ATF6, which is linked with signal transduction. In addition, ENST00000575695.5 primarily regulated UBQLN1, which is linked with hypoxia responses in cells. Previous studies have also explored the roles of G6PC3, ATF6, BMP2, SIX1, RAB7A, and SOX2 in tumors [42–47]

Together our results most clearly highlighted the importance of TEK, LDB2, ATF6 and UBQLN1 in NSCLC patient BE responses, underscoring the value of examining system-level gene interactions in order to better understand the determinants of lung cancer patient therapeutic responsiveness.

Responses to BE treatment may be associated with more favorable patient outcomes. We therefore assessed the relationship between BE response-related hub gene expression and lung cancer patient survival. In public microarray datasets, we found that the elevated expression of TEK, which was upregulated in female BE-responsive patients, was associated with increased survival in female lung cancer patients. Similarly, we found that decreased expression of UBQLN1, which was reduced in male BE-responsive patients, was associated with increased survival in male lung cancer patients. LDB2 and ATF6 were not included in TCGA lung cancer datasets. However, LDB2 has been shown to inhibit liver cancer cell proliferation and migration, functioning as a tumor suppressor [39], while ATF6 has been linked to the induction of intestinal dysbiosis and immune activity associated with the development of colorectal cancer [48]. Reduced ATF6 expression in the BE responders was thus likely associated with better survival.

## Conclusion

In summary, the present study is the first to conduct global mRNA and lncRNA profiling in order to explore sex-related differences in the molecular mechanisms governing NSCLC patient responses to BE therapy. We identified a large number of differentially expressed lncRNAs and mRNAs when comparing datasets from patients that were and were not responsive to BE treatment. Through predictive and functional enrichment analyses, we were able to further uncover potential links between these lncRNAs and mRNAs, offering potential insight into the functional roles of these lncRNAs in the regulation of BE responses in NSCLC. While the present results highlight the unique cancer-associated pathways associated with BE responses in male and female NSCLC patients, more multi-omic studies will be integrated to our present work to discover more possible lncRNA-mRNA correlation in future, and further studies will be needed to validate these findings and to understand the underlying molecular basis for these findings.

## Supporting information

**S1 Fig.**
(TIF)

**S1 Table. GOs intersections.**
(XLSX)

**S2 Table. Pathways intersections.**
(XLSX)

**S3 Table. Nodes in the female lncRNA-mRNA expression correlation network.**
(XLSX)

**S4 Table. Nodes in the male lncRNA-mRNA expression correlation network.**
(XLSX)

**S5 Table. Nodes in the female lncRNA-mRNA-GO-pathway network.**
(XLSX)

**S6 Table. Nodes in the male lncRNA-mRNA-GO-pathway network.**
(XLSX)

**S7 Table.**
(XLSX)

**S8 Table.**
(XLSX)

## Author Contributions

**Conceptualization:** Chao Tu, Yingqi Pi.

**Formal analysis:** Yingqi Pi.

**Project administration:** Chao Tu, Yang Ling.

**Writing – original draft:** Yingqi Pi, Shan Xing.

**Writing – review & editing:** Chao Tu, Shan Xing.

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
