## [Decision Letter · Decision Letter 0]

10 Aug 2020

PONE-D-20-16411

Integrated analysis of long non-coding RNAs and mRNA profiles reveals potential sex-dependent biomarkers of bevacizumab/erlotinib response in advanced lung cancer

PLOS ONE

Dear Dr. Tu,

Thank you for submitting your manuscript to PLOS ONE. After careful consideration, we feel that it has merit but does not fully meet PLOS ONE’s publication criteria as it currently stands. Therefore, we invite you to submit a revised version of the manuscript that addresses the points raised during the review process.

We look forward to receiving your revised manuscript.

Kind regards,

Rama Krishna Kancha

Academic Editor

PLOS ONE

Journal Requirements:

2. Please ensure that you refer to Figure xxxxx in your text as, if accepted, production will need this reference to link the reader to the figure.

Additional Editor Comments (if provided):

The manuscript entitled "Integrated analysis of long non-coding RNAs and mRNA profiles reveals potential sex-dependent biomarkers of bevacizumab/erlotinib response in advanced lung cancer" by Tu et al. demonstrates sex-specific differential expression of biomarkers in relation to drug response in lung cancer. The reviewers are of the opinion that the work is of significant interest. However, the reviewers raised important questions that need to be answered to bring more clarity and highlight the importance of the study.

Reviewers' comments:

Reviewer's Responses to Questions

**Comments to the Author**

1. Is the manuscript technically sound, and do the data support the conclusions?

Reviewer #1: Partly

Reviewer #2: Partly

2. Has the statistical analysis been performed appropriately and rigorously? 

Reviewer #1: Yes

Reviewer #2: Yes

3. Have the authors made all data underlying the findings in their manuscript fully available?

Reviewer #1: Yes

Reviewer #2: Yes

4. Is the manuscript presented in an intelligible fashion and written in standard English?

Reviewer #1: Yes

Reviewer #2: Yes

5. Review Comments to the Author

Reviewer #1: The manuscript sounds 1. non coding genes transcriptional control is elusive and the work provides expression differences by sex dependent markers and 2. lung cancer dependent alterations with noncoding and mRNA profiles by a clinical drug perturbation is elaborated. Therefore, the manuscript is interest to the literature and I recommend the ms.

Reviewer #2: Gist/Summary: The authors come up with identification of several differentially expressed (DE) lncRNAs and mRNAs from publicly available gene expression omnibus (GEO) datasets associated with NSCLC with bevacizumab and erlotinib (BE) treatment. Then they make enrichment/pathway analyses based on gender and identify unique DEs

Strengths: Novel in what they claim that the authors mention on the the first of its kind of work on BE treatment ( which invariably is not so!)

Figures

Weaknesses: The work is based on GEO datasets and not original

Hazy texts in between

Major comments:

Wei et al. 2016 and Saito eta l. 2019, 2020 indeed mention this and identify DE genes ( if NOT lncRNAs) in BE treatment. The authors must crosscheck their works and cite them.

Lines 95-99: The authors must glorify lncRNAs and on why they could be used as biomarkers

Some of the texts go hand-i-hand with clinicaltrials.gov. The authors must check it

Lines 130 in Methods must be renamed. The methodology is out of vigour. As the work is public interpretation of GEO., I wonder why the authors mention about the ethics statement etc., which gives a falsehood to the story.

There must be a methodological flowchart

Have the authors carried out the bidirectional best blast hits whence inferring lncRNAs from BLAST?

The authors must mention whether or not the exon array enrichment they infer uniquely mapped to genes and the lncRNAs were indeed inferred from those genes.

Line 209 on interactions. The combined score of 0.4 doesn't make sense

The NONHSAT095695.2 appears to be largely expressed in liver/adrenal glands NOT lungs. How do the authors justify this?

The whole purpose of making a story for DEs for male and females is not warranted and justified in haste. For example, why males have SLE related DEs is not discussed. The same applies to the coalesce of pathways between males and females.

The multivenn diagram with union of intersection of venns showing 1 and 2 respectively for DE lncRNAs and De mRNAs were not discussed in great detail. For example, where are they localized to and their expression patterns

The conclusions and discussions must herald have point son what if multi-omic studies are integrated to it or say, RNA-Seq expression data is used instead of exon array

Minor comments

The abstract must have an expansion of EGFR

Lines 385 and line 296: Please rewrite or use comma

All figures must be high resolution

6. PLOS authors have the option to publish the peer review history of their article (what does this mean?). If published, this will include your full peer review and any attached files.

Reviewer #1: **Yes: **Yusuf TUTAR

Reviewer #2: **Yes: **Prashanth Suravajhala

---

## [Author Response · Author response to Decision Letter 0]

21 Sep 2020

Dear Editor,

We have now revised our manuscript PONE-D-20-16411 taking into account the useful comments of the academic editor and the two reviewers.

We thank you and the reviewers for your careful reading and the suggestions to improve our initial manuscript.

We took some time to respond to each point raised by the academic editor and reviewers, the detailed responses to the reviewers’ comments are explained below. And we also uploaded our figure files to the Preflight Analysis and Conversion Engine (PACE) digital diagnostic tool (https://pacev2.apexcovantage.com/) to ensure that all of our figures except Supporting Information files meet PLOS requirements. And we checked all themanuscript to make sure that it meets PLOS ONE’s styl requirements, including those for file naming according to the PLOS ONE style templates.

We are confident that these additional modifications strenthened and ameliorated this manuscript.

In addition, the author information was modified according to the contribution of all the authors after the revision.

We hope that the reviewers and yourself will find our manuscript suitable for publication in PLOS ONE.

Best regards,

Chao TU

The detailed responses to the reviewers’ comments are explained below:

Wei et al. 2016 and Saito et al. 2019, 2020 indeed mention this and identify DE genes ( if NOT lncRNAs) in BE treatment. The authors must crosscheck their works and cite them.

We thank the reviewer for providing the references regarding BE treatment. We carefully searched these references, and found that Wei et al. 2018 (PMID:30032815) and Saito et al.2019 (PMID:30975627) did mention about the treatment of erlotinib plus bevacizumab(BE) in non-small cell lung cancer (NSCLC). Both papers investigated the efficacy and safety of erlotinib plus bevacizumab treatment in patients with non-small-cell lung cancer. Instead, the reference of Saito et al.2020 studied the effectiveness and safety of nivolumab treatment, not BE treatment, in NSCLC patients in real-world setting in Japan. However, none of these papers did the study on the profiling of differentially expressed genes or non-coding RNAs after BE treatment in NSCLC patients. Nevertheless, we discussed these papers in the manuscript and cited them as references according to the reviewer’s suggestion.

Lines 95-99: The authors must glorify lncRNAs and on why they could be used as biomarkers.

We agree with the reviewer that we should introduce more about lncRNAs and why they could be used as biomarkers. 

LncRNA is a class of non-coding RNAs of more than 200 nucleotides with biological functions. They are widely involved in biological development and pathological process through chromatin remodelling, cis-regulation at enhancers and post transcriptional regulation. LncRNAs have been found to be specifically expressed in the oncogenesis and metastasis of many tumor tissues (June 2014, Nature Genetics, kiyer). Therefore, these lnRNAs are also considered to be associated with carcinogenesis or tumor inhibition, and are potential biomarkers that can predict the tissue carcinogenesis.

And we added the above introduction concerning why lncRNAs could be used as biomarkers in the manuscript.

Some of the texts go hand-i-hand with clinicaltrials.gov. The authors must check it.

We are sorry that we are not so sure about what the reviewer mentions in this point. We have written the text independent of clinicaltrials.gov. And we did the duplicate checking for our manuscript, and no similar content to clinicaltrials.gov was found.

Lines 130 in Methods must be renamed. The methodology is out of vigour. As the work is public interpretation of GEO., I wonder why the authors mention about the ethics statement etc., which gives a falsehood to the story.

We are sorry for the redundancy of Line 130 in Methods, and it has been renamed to “Data sets”. We agree with the reviewer that it’s not necessary to mention about the ethics statement in Methods as our work is based on the public GEO dataset, therefore, we removed those concerning ethics statement and written informed consent in this part (Please refer to the end of “Data sets” in Materials and Methods).

There must be a methodological flowchart

We have added the flowchart in “Supplementary Figure 1.tif” for the sample selection and data analysis used in the manuscript.

Have the authors carried out the bidirectional best blast hits whence inferring lncRNAs from BLAST?

Yes, we have carried out blast with the bidirectional best hit information method, which is more precise and possesses higher matching degree compared with the single-directional best hit information method.

The authors must mention whether or not the exon array enrichment they infer uniquely mapped to genes and the lncRNAs were indeed inferred from those genes.

We have removed those probes that were not uniquely mapped to genes, therefore, the exon array enrichment we used was uniquely mapped to genes and the lncRNAs were indeed inferred from those genes. We have added this information in Materials and Methods (Please refer to the part of “mRNA and lncRNA profiling” in Materials and Methods).

Line 209 on interactions. The combined score of 0.4 doesn't make sense

We thank the reviewer for this comment. However, we are not so sure about what the reviewer means here. When performing protein-protein interaction (PPI) network analysis, we used the interaction score of 0.4 according to the recommendation of the official developping team for STRING database in this paper from Nucleic Acids Research, 2017:

The STRING database in 2017: quality-controlled protein–protein association networks, made broadly accessible.

And this confidence cutoff of 0.4 is used widely by many other groups, below are some example references that used the interaction score of 0.4 when PPI network:

①Deciphering miRNA transcription factor feed-forward loops to identify drug repurposing candidates for cystic fibrosis. Genome Medicine 2014, 6:94

②Identifification of key pathways and genes in colorectal cancer using bioinformatics analysis. Med Oncol (2016) 33:111. DOI 10.1007/s12032-016-0829-6

③Screening and identification of key biomarkers in hepatocellular carcinoma: Evidence from bioinformatic analysis. 

The NONHSAT095695.2 appears to be largely expressed in liver/adrenal glands NOT lungs. How do the authors justify this?

We thank the reviewer for this comment. In our present study, we only detected the differential expression of NONHSAT095695.2 through bioinformatic analysis by gene annotation, this needs to be validated by experiments. In our future work, we will detect the expression level of NONHSAT095695.2 in lungs of both male and female patients before and after BE treatment by RT-QPCR. 

The whole purpose of making a story for DEs for male and females is not warranted and justified in haste. For example, why males have SLE related DEs is not discussed. The same applies to the coalesce of pathways between males and females.

We thank the reviewer for this point. Gender differences play an important role in the drug response in cancer, especially in lung cancer, for example, in the following papers they all focus their work on sex-dependent drug response in lung cancer:

①Conforti F, Pala L, Bagnardi V, et al. Sex-Based Heterogeneity in Response to Lung Cancer Immunotherapy: A Systematic Review and Meta-Analysis. J Natl Cancer Inst. 2019;111(8):772-781. doi:10.1093/jnci/djz094. 

②Wang S, Zhang J, He Z, Wu K, Liu XS. The predictive power of tumor mutational burden in lung cancer immunotherapy response is influenced by patients' sex. Int J Cancer. 2019;145(10):2840-2849. doi:10.1002/ijc.32327. 

③Tsiouda T, Sardeli C, Porpodis K, et al. Sex Differences and Adverse Effects between Chemotherapy and Immunotherapy for Non-Small Cell Lung Cancer. J Cancer. 2020;11(11):3407-3415. Published 2020 Mar 5. doi:10.7150/jca.40196.

We added the discussion about the coalesce of pathways between males and females in the manuscript. However, we didn't mention SLE-related DEs because no research in our lab is involved in the area of SLE. Sorry for this and thank you for your comprehension. 

The multivenn diagram with union of intersection of venns showing 1 and 2 respectively for DE lncRNAs and De mRNAs were not discussed in great detail. For example, where are they localized to and their expression patterns

Thank you very much for pointing out this issue. We have checked the venn diagram in Fig 5A and Fig 5B, respectively, and found that the union intersection of the significantly enriched two GO terms overlapping the female-upregulated, male-upregulated, female-downregulated and male-downregulated DE mRNA datasets in Fig 5A are : “positive regulation of transcription from RNA polymerase II promoter ”and “negative regulation of transcription from RNA polymerase II promoter”, while the union intersection of the significantly enriched one shared pathway among those four groups in Fig 5B is : “Metabolic pathways”. The detailed mRNAs of the localization in the union intersection of shared GO terms and pathway mentioned above and their expression patterns are provided in Supplementary Table 7 and Supplementary Table 8, respectively. And we have added the discussion about the union of intersection of venns showing 1 and 2 respectively for shared GO functions and pathways in the manuscript according to the reviewer’s suggestion.

The conclusions and discussions must herald have point son what if multi-omic studies are integrated to it or say, RNA-Seq expression data is used instead of exon array

We thank the reviewer for this comment. It is sure that more potential differentially expressed long non-coding RNAs and coding genes can be discovered if RNA-seq data is used instead of exon array data. In future, we will integrate more multi-omic studies to our work in order to discover more possible lncRNA-mRNA correlation. We have added these points in the Discussion and Conclusion part. Please refer to the manuscript for the modifications.

Minor comments

The abstract must have an expansion of EGFR

We have added the full name of EGFR : “Epidermal growth factor receptor” in the abstract. Please refer to the abstract in the manuscript for the modification.

Lines 385 and line 296: Please rewrite or use comma

We have rewritten line 296 and line 385, please refer to the manuscript for the modifications.

All figures must be high resolution.

We have adjusted the figures with low resolution according to the PLOS ONE guidelines: https://journals.plos.org/plosone/s/figures. Now the resolution of all the submitted figures is at least 600dpi. We hope that the newly submitted figures will meet the requirements of PLOS ONE.

---

## [Editor Report · Decision Letter 1]

30 Sep 2020

I ntegrated analysisof long non-coding RNAs and mRNA profiles reveal s  potential sex-dependent biomarkers of bevacizumab/erlotinib response in advanced lung cancer

PONE-D-20-16411R1

Dear Dr. Tu,

We’re pleased to inform you that your manuscript has been judged scientifically suitable for publication and will be formally accepted for publication once it meets all outstanding technical requirements.

Kind regards,

Rama Krishna Kancha

Academic Editor

PLOS ONE
---

## [Editor Report · Acceptance letter]

2 Oct 2020

PONE-D-20-16411R1 

Integrated analysis of long non-coding RNAs and mRNA profiles reveals potential sex-dependent biomarkers of bevacizumab/erlotinib response in advanced lung cancer 

Dear Dr. Tu:

I'm pleased to inform you that your manuscript has been deemed suitable for publication in PLOS ONE. Congratulations! Your manuscript is now with our production department. 

Kind regards, 

on behalf of

Dr. Rama Krishna Kancha 

Academic Editor

PLOS ONE